# Effect of Climate Change Belief and the New Environmental Paradigm (NEP) on Eco-Tourism Attitudes of Tourists: Moderator Role of Green Self-Identity

**DOI:** 10.3390/ijerph20064967

**Published:** 2023-03-11

**Authors:** Abdullah Tarinc, Gozde Seval Ergun, Arif Aytekin, Ali Keles, Ozlem Ozbek, Huseyin Keles, Ozgur Yayla

**Affiliations:** 1Department of Gastronomy and Culinary Arts, Manavgat Tourism Faculty, Akdeniz University, Antalya 07600, Türkiye; 2Department of Tourism Management, Manavgat Tourism Faculty, Akdeniz University, Antalya 07600, Türkiye; 3Department of Social Work, Manavgat Social Sciences and Humanities Faculty, Akdeniz University, Antalya 07600, Türkiye; 4Linda Hotel, Manavgat 07600, Türkiye; 5Gönen Vocational School, Tourism and Hotel Management, Bandırma Onyedi Eylül University, Balıkesir 10900, Türkiye; 6Department of Tourism Guidance, Manavgat Tourism Faculty, Akdeniz University, Antalya 07600, Türkiye; 7Department of Recreation Management, Manavgat Tourism Faculty, Akdeniz University, Antalya 07600, Türkiye

**Keywords:** climate change belief, new environmental paradigm, ecotourism, green self-identity, Alanya

## Abstract

This research has been conducted to determine the effect of tourists’ beliefs of climate change on the NEP and ecotourism attitudes. In addition to this purpose, the moderator role of green self-identity in the effect of the NEP on ecological attitudes has also been examined. The research data were obtained from the tourists visiting the Alanya destination, which is one of the centers that attract the most tourists in Turkey. When the results of the research were examined, it could be determined that the belief in climate change is effective on all dimensions of the NEP, and similarly, all dimensions of the NEP have also affected the tourists’ ecological attitude. Further, green self-identity has a moderator role in the effect of ecocentric and anthropocentric sub-dimensions on eco-tourism attitudes. As a consequence of the findings, a number of theoretical and practical implications have been developed for sector managers, destination management organizations, and academicians.

## 1. Introduction

The “environmental” movement emerged in the United States in the late 1960s due to the fact that people have interacted with the environment throughout history. The understanding that is necessary to increase awareness of protection and have a balanced system in human and nature interactions was defined and programmed in the Brundtland Report published in 1987. In the report, sustainable development is defined as the process of “meeting the needs of the present without harming the needs of future generations, with a balance of conservation and utilization”. This definition includes two different approaches. The first is that the anthropocentric approach focuses on meeting the basic needs of human beings today and in the future. As for the second approach, the ecocentric approach, it focuses on environmental balance. Its primary aim is to balance and protect environmental values to the extent that they can meet the needs of humans both today and in the future [1]. This issue was studied in the 1970s with the “primitive beliefs” theory, which examines the basis of individual attitudes in response to environmental problems and examines cause and effect in the interaction between humans and the world [2]. In forming the New Environmental Paradigm (NEP), Dunlap has addressed the issues of “ecological limits, the importance of preserving nature’s balance, and the rejection of a human-centered belief that nature exists for human use” [3]. In addition to the effect of political ideologies, cultural values, and demographic characteristics of beliefs about climate change, many studies have also shown a positive relationship between the experience of extreme weather events and concern about climate change [4,5]. In these studies, extreme weather experiences have been found to provide a more realistic and localized sense of climate change. Due to this perception of reality, the cognitive barriers and concerns regarding climate change action have been reduced [6,7]. However, some other studies have not been able to reach clear conclusions on extreme weather events; in these studies, the perceived risk of climate change has been found to be associated with the desire to adapt to climate [8,9]. When the studies have been examined, it can be stated that climate change is a global phenomenon with significant effects on societies [10], economies [11], and ecosystems [12].

The tourism industry is particularly vulnerable to the impacts of climate change as it is closely connected to the natural environment and is dependent on its protection [13]. Eco-tourism, which involves travel to natural areas with a focus on environmental protection and education, has emerged as a response to the environmental challenges posed by traditional tourism. It is crucial to understand the attitudes and beliefs of tourists concerning climate change and the environment in order to promote eco-tourism and develop sustainable tourism practices. Previous studies have revealed a positive relationship between environmental attitudes and eco-tourism behavior [14,15,16]. However, there is no research on the variables and factors that affect the attitudes and behaviors of tourists in terms of eco-tourism.

This study examined the relationship between two key variables, climate change belief and NEP, and their effect on eco-tourism attitudes of tourists. Climate change belief refers to the extent to which individuals perceive climate change as a real and urgent problem, while NEP reflects individuals’ attitudes regarding the environment and their motivation to take action to protect it [3,17]. Moreover, the study also investigated the moderator role of green self-identity in the relationship between NEP and eco-tourism attitudes. The study aims to provide insights into the factors that influence eco-tourism attitudes and behaviors of tourists by exploring the complex interaction between the aforementioned variables in light of data obtained from the tourists staying in five-star hotels in the Antalya/Alanya district, one of the most important destinations in Turkey. The findings of this study can inform the design of effective communication and marketing strategies that promote eco-tourism and encourage tourists to engage in environmentally friendly behaviors. Furthermore, the study can contribute to the development of more sustainable tourism destinations and attractions that are essential for the conservation of natural ecosystems and biodiversity.

## 2. Conceptional Framework

### 2.1. Climate Change Belief and New Environmental Paradigm

Awareness of the existence of climate change is an important issue for all individuals. While individuals living around the world face the threat of climate change, regardless of demographics, having an awareness about the issue has now become a matter of responsibility for everyone. In particular, acting with the awareness of the threats posed by climate change is considered a routine behavior that should be present in the lives of everyone around the world. In addition to being knowledgeable and educated on the subject, it becomes more important that individuals take responsibility actively in the process [18,19]. Even though there are important evaluations and directives for all individuals worldwide to be conscious and aware, it is possible to state that the people who are directly affected by the changes related to the environment and nature act more consciously and are more knowledgeable and concerned about this issue.

On the other hand, it is noteworthy that there are different approaches to climate change belief. Climate change beliefs, which have different qualities within the understanding of individual, corporate, and civil society, cause different opinions on the harmful effects of this event. While some people believe that climate change is an event that takes place within the structure of nature, for some, the problem is largely due to the way of life of human beings. It is also observed that there are individual and social demographic differences in climate change belief. Nevertheless, sudden and extreme changes in air temperatures are effective in reshaping individuals’ views on climate change [20,21]. Anthropocentrism is an approach that envisages the mostly human-based treatment of earthly values. In this approach, while humans are the main factor, the view that other beings in the world are created for the interests and expectations of people comes to the fore. Anthropocentrism, ethically, has been criticized by many. The main reason for this is they believe that a people-oriented approach eliminates other values in the world. The view foregrounded within anthropocentrism is the perceived necessity of a distinction between legitimate and illegitimate interests of people. However, in this approach, keeping the expectations of individuals above the expectations of environmental factors is considered a factor that disrupts the balance [22,23].

The concept of anthropocentrism is often criticized for emphasizing human supremacy over the world. The basis of this criticism is the view that human power and rights are exaggerated. While the basis of sociological and environmental problems around the world is seen as human, anthropocentrism and the glorification of human beings or placing humans in the center are generally heavily criticized. The general view on anthropocentrism is that it evaluates the system with a single element [24,25].

Ecocentrism is an approach that has been developed for many years to eliminate the human-centered view of the world and life that people have. With ecocentrism, the importance of the world outside the human being increases, and in this way, environment-based and eco-centered approaches that emphasize all elements and life factors in the world are adopted. Ecocentrism is a perspective that thinks that non-human life elements have a direct or indirect effect on human life and therefore moves away from seeing man as the singular entity of the universe. At the same time, ecocentrism is considered an important element of social education. Accordingly, ecocentrism is called a valuable element for social development and consciousness within national and international education mechanisms [26,27]. 

The concept of ecocentrism, which is much more ethically prioritized, is used to make sustainable environmental understanding acceptable to all societies and individuals, regardless of whether it is local or international. In the past years, sociologically, the human-based understanding of ethics has been considered much more important. However, in the current process, an environment-based ethical understanding has been added to the human-based ethical understanding, and this understanding helps the ethics concept to gain a more comprehensive and universally stronger common character. In fact, lately, it has been seen that the process has been handled collectively, with the articulation of environmental rights, as well as human rights [23,28].

Dietz, Dan, and Shwom [29] found that the research participants had a high level of awareness of natural life problems that emerged with climate change in the research they conducted in the states of Michigan and Virginia. According to the findings of the researchers, in parallel with the NEP, the participants moved away from the understanding that ignores environmental factors. Again, respondents lost confidence in industrial companies and governments regarding environmental issues, while they placed more trust in environmental NGOs. 

Ntanos et al. [30], in their research on Greece, focused on the increasing environmental awareness with the NEP and the prominent issues related to climate change. The answers given by the participants in the research show that environmental awareness spreads rapidly among individuals. Climate-based change and emerging threats have a significant impact on this. In addition, participants tend to learn about and use renewable energy sources.

Prober et al. [31], in their research based on literature on the subject, identified an environmental awareness approach that has been gaining momentum since 2008. Accordingly, studies in the literature on the subject show that, as a result of the climate change factor becoming a serious threat, there have been radical changes in environmental assessments, and this change has helped to create a new environmental life-building awareness in society. 

Derdowski et al. [32], in their research on the subject, determined that there is a significant and positive relationship between the NEP and the development of environmental consciousness of individuals. According to the findings of the researchers, the NEP shows that individuals evaluate seasonal and climate-based changes negatively, and therefore, they approach environmental issues in a more detailed, more careful, and more analytical way. 

Diakakis, Skordoulis, and Savvidou [33] investigated the experiences and changes in opinions of people about climate change and environmental awareness, according to the picture that emerged as a result of natural disasters in Greece. Accordingly, the participants of the research have better grasped the importance of environmental issues as a result of the natural disasters they have experienced or encountered, and they continue their lives with environmental awareness after the events they have experienced. In particular, as the degree of impact of natural disasters increased, the environmental awareness, practice, and approach sensitivities of individuals after the disaster increased rapidly.

It has been observed in previous studies (see Table 1) that individuals who were exposed to natural disasters, air pollution, and climate change are more likely to be environmentally conscious. Regarding environmental sensitivity, it is noteworthy that there is a decrease in trust in industrial companies and governments, but an increase in environmental NGOs. A significant and positive relationship among the new environmental paradigm, increased environmental awareness, and climate change has been clearly identified through the research. Within the scope of the literature, the following hypotheses have been formed:

**H_1_.** 
*There is a positive and significant effect of climate change beliefs on NEP.*


**H_1a_.** 
*There is a positive and significant relationship between climate change beliefs and ecocentric views.*


**H_1b_.** 
*There is a positive and significant relationship between climate change beliefs and balanced growth.*


**H_1c_.** 
*There is a positive and significant relationship between climate change beliefs and anthropocentric views.*


### 2.2. New Environmental Paradigm and Eco-Tourism Attitude

Dunlap and Liere [3] developed the NEP scale to illuminate the new world view towards environmental attitudes. The NEP reveals attitudes about the inevitability of limitations to growth, the need to achieve a sustainable economy, the importance of protecting nature, and the necessity to reject the anthropocentric view that nature exists only for human use. The NEP has been widely used in environmental issues since 1978 [3,37]. 

According to Dunlap and Van Liere [3], while global realities, such as technology, property dependency, and continuous growth that make social expectations shine more, point to a dominant social paradigm, this understanding also becomes the intellectual ground of environmental degradation. For this reason, there is a transition from anthropocentric thought that puts the human in the center to an ecocentric approach, that is, an NEP, in which the balance of nature is observed [2,38].

The NEP, which is considered more as a scale in the literature, helps individuals to evaluate their views on ecological borders, the balance of nature, the dominance of humans over nature, and ecological disasters. The scale mostly includes statements on subjects, such as the importance that individuals attach to environmental problems, whether they have a human-centered or nature-centered understanding, and how they establish a relationship between technology and environmental problems [30,39].

Individuals who adopt this paradigm generally perceive the state of the world as a problem, believe that there are limits to growth that people must adapt to, and argue that the NEP is a set of beliefs that positively affect behavior and attitudes towards the environment. While the NEP scale is discussed in the literature in areas, such as education, agriculture, and tourism, it has been observed that there are limited studies on outdoor recreational activities [40,41].

Ecotourism is the abbreviation of ecological tourism. Ecotourism refers to a tourism approach that does not spoil and preserves nature. This kind of nature-preserving understanding emerged in the 1980s. Accelerating developments in the world, the negative effects of tourism activities on natural and cultural resources and the understanding that these effects endanger tourism’s own future have brought tourism types based on the longer-term use of resources to the agenda [42,43].

Although the definition of ecotourism has been made in different ways by many scholars, in 1996, the International Union for Conservation of Nature (IUCN) defined ecotourism as benefiting local people, enabling active socio-economic participation of the public, creating a low level of negative visitor impact, and improving conservation (past and present cultural riches) as a concept of tourism. At the same time, the IUCN has defined ecotourism as environmentally responsible travel and visitation to relatively unspoiled natural areas in order to appreciate and enjoy nature [44,45]. 

In all the old and new definitions of ecotourism, more specific interests and purposes, such as the desire to return to nature, the desire to experience nature, the desire to escape from the pressures of daily life, and the desire to see nature before it disappears, are given as the reason why the activity is nature-oriented. Meanwhile, the travel and tourism industry’s market expansion objective is either not communicated or presented as a service. The common definition of ecotourism has become more complex over time, enriched by sustainable tourism and alternative tourism views. In these new definitions, the aims and outcomes of ecotourism are determined as funding for environmental protection, scientific research, protecting untouched and sensitive ecosystems, benefiting rural people, and promoting development in poor countries. At the same time, there are activities to develop ecological and cultural awareness, to develop social responsibility in the tourism industry, to increase environmental awareness, to educate tourists, and to support world peace [46,47]. The studies on the subject in the literature have been summarized in Table 2.

The previous studies have used the NEP scale to measure environmental or eco-tourism attitudes in a number of destinations involving different nationalities and groups of participants. In line with the previous research, the present study has the purpose of determining the relationship between eco-tourism and NEP in the Alanya destination. Based on the findings in the literature and the table above, the following hypotheses have been developed:

**H_2_.** 
*There is a positive and significant effect of NEP on eco-tourism attitudes.*


**H_2a_.** 
*There is a positive and significant relationship between ecocentric views and eco-tourism attitudes.*


**H_2b_.** 
*There is a positive and significant relationship between balanced Growth and eco-tourism attitudes.*


**H_2c_.** 
*There is a positive and significant relationship between anthropocentric views and eco-tourism attitudes.*


### 2.3. Moderator Effect of Green Self-Identity on the New Environmental Paradigm and Eco-Tourism Attitudes

Green self-identity is an important motivation for an individual to identify with the green consumer in general within the scope of ecotourism activities and to acquire behaviors and attitudes to protect and sustain nature by focusing on the environment. Green self-identity, which is an issue that individuals with high environmental awareness attach importance, is evaluated in relation to recycling and sustainability issues. Individuals, as consumers, can shape their consumption habits in this direction if their green self-identity awareness is high [55].

On the other hand, green self-identity creates a consciousness and responsibility in terms of consumers. Individuals evaluate everything within green elements. This kind of evaluation tendency creates the situation that the issue should be handled with ethical dimensions. In particular, it is seen that a consumption understanding based on ethical elements has become established [56]. Luzar et al. [57] examined the views of tourists in the US state of Louisiana on the change in touristic activities in parallel with the NEP. Accordingly, tourists in the state expect touristic businesses to consider much more environmental elements now. On the basis of this expectation, the increasing environmental awareness of tourists and the fact that the subject has gained popularity throughout the state has a significant impact. In addition, tourists have the view that an environmentally friendly tourist business can offer a more exotic and different life experience. 

The phenomenon of ecotourism, arising from an understanding based on environmental protection, has been defined as travel to natural destinations while protecting the welfare of local people. As ecotourism awareness develops, nature education, environmental and cultural protection, human rights, and ethical values have been included in the concept [58]. The experience of interacting with the unspoiled nature and culture of the region is indispensable for ecotourism when traveling to ecological destinations with small tourist groups. With the spread of ecotourism and its adoption by tourists, it is possible to observe the wide-ranging effects of this phenomenon on tourist behavior and the management organization structure of touristic enterprises in green self-identity comprehensive studies. It is expected that the concept of green self-identity, which includes details about the way of interacting with the natural environment, will have a positive effect on ecotourism behaviors. This is because ecotourism behaviors consist of the components of the behavioral intentions of the tourists at the end of the consumption process [59].

Deng, Walker, and Swinnerton [60] examined the differences in the eigenvalues determined by individuals in terms of NEP behaviors in terms of Chinese and Anglo-Canadian participants. While the participants think differently about the limits of growth and anti-anthropocentrism, they think parallel to each other in terms of their concerns about eco-crisis and nature balance. Accordingly, while the participants determine their views on the axis of the NEP, they also shape their own practices and activities regarding environmental issues in this direction.

Luo and Deng [61] investigated the direction and form of touristic preferences and behaviors of tourists in the axis of the NEP. The most important finding of the researchers was that the participants are now more carefully in search of environmental elements in their touristic activities. In this way, beyond being valuable as a customer, they expect the environmental elements from which they benefit to be considered much more valuable. In this way, participants wish to reduce the level of consumption incentive-oriented activities of touristic areas and businesses.

Zhang et al. [62] investigated the effects of shaping individuals’ views on environmental issues and their effects on touristic preferences. The most valuable finding of the study is that the participants’ encounters with natural disasters have an extremely important effect on determining their environmental awareness. Beyond that, the researchers stated that this situation is also effective in shaping the tourism activities along with the life preferences of the participants. At the same time, the researchers suggest that individuals’ views and environmental perceptions should be taken into account in shaping touristic areas. 

Li and Wang [15] conducted research on the effect of eco-tourism visitors’ environmental attitudes on environmental behavior. According to the findings of the researchers, there is a direct link between the satisfaction of tourists and their ecotourism preferences. Accordingly, the participants carefully examine the sensitivities of touristic areas and touristic businesses based on environmental issues during their touristic visits. This situation not only affects the preferences of tourists, but also helps them to develop their views on ecotourism. 

Lorenzo-Romero, Alarcón-del-Amo, and Crespo-Jareño [63] compared the approaches of Chilean and Spanish tourists to ecotourism. According to the findings of the researchers, there are two issues that come to the fore in the determination of the decisions taken by the tourists about ecotourism. The first of these is the cultural tendencies of tourists in this regard. Cultural values and innovative perceptions of tourists about environmental issues are effective in shaping their decisions on ecotourism. The second point is the experiences of the participants based on ecotourism in the previous period. In this regard, the experiences in the past years encourage new steps to be taken in the field of ecotourism. 

Zhang et al. [64] investigated the effect of cultural heritage on individuals’ preferences for touristic activities and its change over time. According to the findings of the researchers, the perceptions of consumers about tourism change over time. While not so intense, consumers are increasingly looking for sustainable and environmentally friendly tourist attractions and services.

Nowacki, Chawla, and Kowalczyk-Anioł [65] investigated the factors that shape the eco-friendly tourism preferences of tourists with the specific of Indian participants. Accordingly, the existence of an environmentally friendly destination and the social and personal norms and behavioral patterns determined individually by the participants have an important place in determining the direction of their touristic activities. At the same time, participants are more willing to travel to ecologically protected destinations. Young and highly educated tourists participating in the research are more willing to accept and implement ecological touristic activities in a sustainable way. At this point, the participants do not consider any financial constraints in visiting such ecological destinations. 

Jeong et al. [52] investigated the relationship between a national park experience and tourists’ understanding of ecotourism. According to the evaluations of the researchers, the participants show that with the national park experience, the elements of nature become more remarkable and the participants’ awareness of the importance of natural areas increases. In addition, the intellectual climate created by the national park experiences in order to shape the preferences of the participants on ecotourism has a guiding quality for touristic activities.

When Table 3 has been examined, it is observed that there are not that many studies on the concept of green self-identity, and moreover, a study on the moderator role in the relationship between NEP and eco-tourism variables has not been encountered. Based on this situation, H_3_, which constitutes the primary unique aspect of the study, has been formed.

**H_3_.** 
*Green self-identity has a moderator role in the relationship between NEP and eco-tourism attitudes.*


**H_3a_.** 
*Green self-identity has a moderator role in the relationship between ecocentric views and eco-tourism attitudes.*


**H_3b_.** 
*Green self-identity has a moderator role in the relationship between balanced growth and eco-tourism attitudes.*


**H_3c_.** *Green self-identity has a moderator role in the relationship between anthropocentric views and eco-tourism attitudes. Thus, based on the literature review, the proposed model has been presented in Figure 1*. 

## 3. Materials and Methods

### 3.1. Research Instrument 

The scales, which were considered in accordance with the research purpose, were developed through a comprehensive literature review [3,70,72,76,77,78,79,80,81,82,83,84,85,86,87,88]. The questionnaire used in this context consists of four parts. The first part of the scale, which consists of 7 statements, regarding the climate change belief (CCB), was prepared by Arbuckle et al. [79]. The scale concerning NEP, which consists of 12 statements under 3 dimensions, was adapted from the study conducted by Dunlap and Van Liere [76]. In addition, 3 statements about green self-identity were taken from the study of Bradley et al. [84]. Finally, 6 expressions of the eco-tourism attitude scale were based on the research carried out by Castellanos-Verdugo et al. [88]. Due to the fact that the participants could be of different nationalities within the research sample, the original scales were translated from English into Russian and German. At this stage of the research, two different language experts for both languages were used applying the back-to-translation method [89] to translate the statements from English into Russian and German. First of all, 28 statements in the questionnaire were translated from English to Russian and German, and then, the same statements were translated into English by the second language expert, and thus, they were double-checked. Then, the questionnaire form to be used in the research was applied to the academic staff working in the field of tourism, and it took its final form considering their recommendations. The statements used in the study were scored on a five-point Likert scale ranging from 1 (strongly disagree) to 5 (strongly agree). In addition to these, the questionnaire form included questions about the age, gender, marital status, and education level of the participants in order to determine their demographic characteristics.

### 3.2. Sampling and Data Collection

The research was carried out in Alanya district of Antalya province, which contributes significantly to Turkey’s tourism revenues. Alanya is considered one of the important tourism destinations of Antalya along with Belek, Manavgat, and Kemer. Although it is 135 km from the Antalya city center, Alanya is a developed tourism destination in terms of accessibility with many connection roads and the airport that belongs to the region. Tourism constitutes a large part of the destination’s economy through the mass tourism activities particularly within the context of sea, sand, and sun tourism. According to the 2019 report released by the Alanya Chamber of Commerce and Industry [90], in 2019, approximately 6.7 million domestic and foreign tourists visited the destination during the high tourism season before the COVID-19 pandemic. According to the data taken from the Ministry of Culture and Tourism [91], there were 280 accommodation establishments with tourism operation certificates in Alanya in 2021 and 92 of them are five-star hotel establishments. Further, when the number of hotels is evaluated, it can be understood that one out of every three hotels in Alanya is a five-star hotel [91]. While Alanya has a total bed capacity of 125.031 beds, 58.093 beds are in five-star hotels. This result indicates that 47% of the bed capacity in Alanya has been provided only by five-star hotels. While this share is 28% in four-star hotels, it is 7% in three-star hotels [92]. The remaining 18% of the bed capacity has been provided by other accommodation businesses. Therefore, based on the criteria and characteristics mentioned above, it was decided to conduct the research in the Antalya/Alanya destination. As a popular tourism destination, Alanya has provided significant economic contributions to both the country and the local people by meeting 7% of the total bed capacity of the country, hosting 10% of the foreign tourists, and constituting 13% of the total tourism revenues [90,91]. 

In addition to these, Alanya has 0.5 per cent of Turkey’s forest assets. Due to the plant diversity, trekking and hiking themed tours are in demand in the region, and also hunting tourism is of interest in the Cebireis and Akdağ region. Along with ecotourism activities, there are also many caves in the region, which could be considered destinations for health tourism-themed tours. In this regard, Damlataş Cave can be given as an example [93]. The ecotourism supply of the region is protected by various projects in line with the sustainability policy of the local administration. The Sapadere Canyon Ecotourism Area and Trekking Trail has been the center of attention for daily activities of local and foreign tour organizations in Alanya. The tourists enrich their tour experiences with nature sports, such as hiking, cycling, and photography in the area [94].

It is stated that the global ecological footprint (EF) in mass tourism is higher than that in nature-based tourism types, such as ecotourism [95]. Furthermore, the studies have found that the airline preference in travelling leads to high carbon emissions [96,97,98]. Only from international flights, tourists are intensively transferred to Alanya from Antalya, which is the second airport with the highest passenger traffic in Turkey with 25,130,857 passenger registrations as of 2022. Likewise, Gazipaşa airport, with 273,402 international passengers as of 2022, also provides Alanya with a large number of visitors [99]. In this context, it is clear that the airline is frequently preferred in transportation to the Alanya destination. In addition, considering the accommodation statistics of Alanya, five-star hotel bed capacities [92] also indicate the intensity of mass tourism. Based on this fact, mass tourism and intensive airline use can be considered an indicator of high EF in Alanya. As a matter of fact, this research approached tourism from an environmental perspective. 

In the light of all these data, it can be inferred that Alanya, which provides mutual benefits for both the local people and tourists, is quite a developed destination regarding the tourism sector. Moreover, the total carrying capacity of the destination is generally exceeded in the summer season when the tourism activities are intense. It can be stated that the population of the district is much higher than it should be, and thus, tourism activities have many positive and negative effects on the environment. For these reasons, Alanya was chosen as the research population.

According to Becken et al. [100], it was found that, based on the accommodation category of tourists, (hotel, B&B, motel, backpacker, campground, home), the highest energy consumption per person per day was in the hotel category (155 MJ/visitor night). Pieri and Santamouris [101] classified hotels according to their size with 8 criteria as cluster 1 (average 319 beds), cluster 2 (average 510 beds), and cluster 3 (average 121 beds), and although cluster 2 has the lowest consumption in kw h/m2 (cluster 1, 2, and 3, 3.18 kw h/m^2^, 2.82 kw h/m^2^, and 8.7 kw h/m^2^), as Idahosa et al. [102] and Tsai et al. [103] state, when the number of rooms in a hotel business increases, the amount of electricity consumption will also increase. As for the star rating category, it has been determined that there is no significant difference between the energy consumption of three and four-star hotels, but five star hotels have significantly higher electricity consumption. This is supported by previous research that all of the hotels selected in the study have a capacity of more than 500 beds, which is an important sample in terms of energy consumption and environmental impact. Additionally, Mensah [104] found that as the number of stars increases, the environmental management performance of hotels increases in the areas of “environmental health and pollution prevention”, “green purchasing”, “eco labelling and certification”, “water and energy efficiency”, and “environmental education to guests and staff”. When the literature is examined, it has been observed that as the number of stars of the hotels increases, the amount of resource utilization and environmental management performance increases as well, and therefore, it is thought that conducting the research on five-star hotels may yield much more significant results.

A survey was conducted on the guests staying in five-star hotels within the context of the research. The reason for applying the survey only to the guests of five-star hotels is the assumption that more samples could be reached thanks to their larger size compared to other hotel businesses. Firstly, an e-mail was sent to 92 five-star hotels about the content of the research and the target group. At the end of one week, 9 hotels responded positively to the implementation of the research questionnaire. The visitors of 9 five-star hotels, who accepted to participate in the research, located in Avsallar, Okurcalar, Türkler, and Mahmutlar districts of Alanya, constituted the sample of the research. First of all, in the first week of June 2022, the pilot study was conducted by applying the questionnaires to 48 foreign visitors staying in 2 hotels within the sample. The reliability analysis and structure comprehensibility of the scales were checked through the data obtained, and it was determined that there was no problem in the analysis results, and thus, the process of collecting the actual research data was started.

The research data were collected through the convenience sampling method, one of the simple random sampling methods, and partially via the snowball sampling method. Limiting the research area to a specific sample may weaken the representative power of the sample [105]. At this point, researchers suggest limiting the number of questionnaires to be taken from a unit [106]. Considering this information, a maximum of 60 questionnaires was collected from a hotel in the study, and thus, the study aimed to increase the generalizability of the findings. In the creation and implementation process of the questionnaire special attention was paid to the common method bias, which was emphasized in the study carried out by Podsakoff et al. [107], and a number of response development techniques were applied to minimize possible errors. For each questionnaire form, a cover page was prepared with information, such as “Participation is optional”, “The data collected during this research will be kept confidential”, and “There is no right or wrong answer in this survey” [108]. A total of 458 questionnaires were applied and collected in the third and fourth weeks of June via the voluntary participation of foreign visitors staying in the hotels within the sample of the dataset. The research analyses were carried out with the remaining 434 questionnaires after excluding the incomplete and incorrectly filled questionnaires.

### 3.3. Data Analysis

It is stated in literature that when it is not possible to reach the population, 5 to 10 times the number of items used in the scale will be sufficient as the sample for the research [109]. There are 28 items in the scale, and it is clear that the number of samples (434) was 15 times the number of statements (28). On the other hand, at a 95% confidence level, at least 370 questionnaires should be included in the analysis in cases where the population size is unknown or more than 10,000 [110]. In this context, it is thought that the sample was sufficient to represent the population. The obtained dataset was transferred to the SPSS Statistics Base V23 program. Before analyzing the relationships among dependent, independent, and moderator variables, data scanning was performed. In this context, firstly, data scanning was performed, and then, Mahalanobis distance was analyzed to determine the extreme values. As a result of the data-scanning process, 9 questionnaires were excluded from the analysis since they contained extreme values (Mahalanobis’ D(27) > 0.001). 

In the second stage, the multicollinearity problem was evaluated. As a consequence of the analysis, it was determined that the VIF values were below 5 and the tolerance values were above 0.10. In light of this evidence, it was concluded that there was no multicollinearity problem [111]. In the last stage, the normality distribution of the data was analyzed, and it was determined that the kurtosis and skewness values of the expressions were between −1.5 and +1.5. As a result of these results, it was concluded that the data showed a normal distribution [112]. Based on this finding, the AMOS program was used to test the structural model developed for the purpose of the research. In the H3a-b-c hypotheses included in the research model, the moderator role of green self-identity in the relationship between the sub-dimensions of NEP and the eco-tourism attitude was examined. The Process Macro statistical program developed by Hayes [113] was used for analysis. Model 1 was selected in the Process Macro statistical program for moderator effect analysis.

## 4. Results 

### 4.1. Demographic Profile

Of tourists who answered the survey questionnaire, 65.6% were female (*n* = 279) and 34.4% were male (*n* = 146), and 26.8% of the participants were 18–25 (*n* = 114), 37.9% were 26–34 (*n* = 161), 27.8% were 35–44 (*n* = 118), 6.1% were 45–54 (*n*= 26), and 1.4% were between 55-years-old and over (*n* = 6). When their marital status was examined, it was determined that 32% of the tourists were married (*n* = 136), and 68% were single (*n* = 289). Finally, when the education levels are examined, 1.6% of the tourists listed primary school (*n* = 7), 6.4% listed secondary school (*n* = 27), 11.8% listed an associate degree (*n* = 27), 72.7% listed undergraduate (*n* = 309), and 7.5 of them had a postgraduate education (*n* = 32).

### 4.2. Confirmatory Factor Analysis Regarding the Structural Model

Before testing the research model, the process recommended in the literature was followed [114]. In this context, confirmatory factor analysis was first applied to the data obtained, and the extent to which the collected data overlapped with the model was tested. The results have been presented in Table 4. One of the first values to be examined within the scope of confirmatory factor analysis is factor loadings [115]. It can be seen that the factor loadings of all statements in the scale had a value of 0.50 and above. In addition, the t values calculated for all statements were found to be significant (*p* ≤ 0.001). When the goodness of fit values of the model were examined, it could be stated that the relevant values were within acceptable limits (χ^2^ = 665.083, df = 335, χ^2^/df = 1.955, NFI = 0.928, IFI = 0.964, TLI = 0.959, RFI = 0.919, RMSEA = 0.047, CFI = 0.963).

On the other hand, the reliability, convergent validity, and composite reliability values of each construct were examined. At this point, it was decided that each construct met the reliability condition, as the Cronbach alpha values for all constructs were 0.70 and above [116]. The minimum CR value for each latent variable was found to be 0.838, and the minimum AVE value was found to be 0.605. These results indicate that the study meets the convergent validity and composite reliability values [111]. In light of the satisfactory results obtained, the process of testing the model was started.

### 4.3. Hypothesis Testing

In parallel with the confirmatory factor analysis, the results of the goodness of fit values obtained from the path analysis were also at an acceptable level (χ^2^ = 715.409, df = 269, χ^2^/df = 2.660, NFI = 0.911, IFI = 0.943, TLI = 0.936, RFI = 0.901, RMSEA = 0.063, CFI = 0.943). When the hypothesis results were evaluated, it could be stated that belief in climate change positively affects all sub-dimensions of the NEP. Within the context of the results obtained, a one-unit change in belief in climate change increases ecocentric views by 19% (β = 0.19, t = 3.628, *p* < 0.001), balanced growth by 15% (β = 0.15, t = 2.900, *p* < 0.05), and anthropocentric views by 14% (β = 0.14, t = 2.571, *p* < 0.05). In light of these results, H_1a_, H_1b_, and H_1c_ were accepted. Similarly, the sub-dimensions of the NEP, ecocentric views (β = 0.49, t = 8.066, *p* < 0.001), balanced growth (β = 0.35, t = 6.394, *p* < 0.001), and anthropocentric views (β = 0.41, t = 7.258, *p* < 0.001) have positively affected ecotourism attitudes. Based on these results, H_2a_, H_2b_, and H_2c_ hypotheses were accepted.

### 4.4. Moderator Effect Results

The results of the regression model established in order to test the hypotheses developed for the moderator effect of the research are presented in Table 5. When the table is examined, it was determined that the moderating role of green self-identity in the effect of ecocentric views on ecological attitudes is significant (β = 0.08, 95% CI [0.001, 0.156], *p* < 0.05). According to this result, H_3a_ was accepted. Moreover, it was found that the effect of ecocentric views on ecological attitudes is weaker for individuals with low green self-identity (β = 0.49, 95% CI [0.368, 0.624], *p* < 0.001), while the effect is stronger for individuals with high green self-identity (β = 0.65, 95% CI [0.524, 0.784], *p* < 0.001) as a consequence of a detailed analysis concerning the aforementioned moderator effect. Similarly, green self-identity was found to play a moderating role in the effect of anthropocentric views, another sub-dimension of the NEP, on ecological attitudes (β = 0.07, 95% CI [0.008, 0.129], *p* < 0.05). Therefore, H_3c_ was accepted. Moreover, while the effect of anthropocentric views on ecological attitudes was weaker for individuals with low green self-identity (β = 0.19, 95% CI [0.104, 0.295], *p* < 0.001), the effect was stronger for individuals with high green self-identity (β = 0.33, 95% CI [0.232, 0.444], *p* < 0.001). On the other hand, the moderating role of green self-identity in the effect of balanced growth on ecological attitudes was tested, and as a result of the evaluation, the moderator effect was found to be insignificant (*p* > 0.05). For this reason, H_3b_ was rejected.

## 5. Conclusions

The aim of this study was to determine the effects of individuals’ attitudes concerning climate change beliefs, the NEP, and attitudes towards ecotourism on each other and to enable an understanding of the moderator role of green self-identity on the basis of the climate crisis, on which many effects have been discussed in recent years. The results of the study have shown that climate change beliefs have a positive effect on the dimensions of the NEP, which are ecocentric, balanced growth, and anthropocentric approaches. In this sense, an increase in climate change beliefs has a positive effect on the NEP. These results are consistent with the studies in literature. In a study conducted on Mandarin-speaking individuals in Beijing, China by Xue et al. [117], the authors found a positive relationship between global warming risk perception and the sub-dimensions of the NEP, specifically ecocentrism, and a negative relationship with anthropocentrism. Even though there are studies in the literature that have shown a negative effect of anthropocentrism [22,117,118], in our research, a positive effect was observed. The reason for this effect to have been positive could be explained by the fact that the statements of anthropocentrism included in the scale used to measure the sub-dimensions of NEP are positively oriented. The results of the hypotheses are given in Figure 2.

Another important result that was obtained from the model is the effect of NEP dimensions on ecotourism attitudes. According to this, as the ecocentric approach, balanced growth, and anthropocentric approach increase, the ecotourism attitude also increases. According to Luo and Dang [61], those who support the sub-dimension of limited growth within the NEP have a higher nature-based tourism motivation than that with other dimensions. In the research carried out by Luzar et al. [57], they found that the NEP-based scale is a significant explanatory factor in ecotourism participation. In this scope, the results obtained from our study have presented parallelism with the results of previous studies. In addition, the moderating effect of green self-identity on the effect of the NEP dimensions on ecotourism attitudes has been partially supported. It was found that green self-identity plays a moderating role in the effect of ecocentric and anthropocentric approaches on ecotourism attitudes. To clarify, it can be observed that the intensity of the effect between the ecocentric and anthropocentric approach and ecotourism attitudes is affected by the increase and decrease in green self-identity.

Moreover, through the analysis conducted to measure the moderator role of green self-identity in the effect of balanced growth on ecotourism attitudes, no effect was detected. Accordingly, it was defined that green self-identity is not important for the effect of balanced growth on ecotourism attitudes.

## 6. Discussion

This research was carried out in order to determine the relationships between variables that are of great importance from the perspective of sustainability, in Alanya district of Antalya, one of Turkey’s most popular tourist destinations. Alanya is one of the important tourism centers, which has received approximately 13 million visitors from 2022 [93]. The data obtained from the research provide some theoretical and practical contributions to tourism workers, sector representatives, and academics working in the literature.

### 6.1. Theoretical Implications

When the findings obtained as a result of the analysis made in the research have been evaluated, a number of remarkable results have been revealed. It is thought that this research will provide added-value to the literature, especially with the comments to be made on the findings obtained in the moderator effect tests. In this context, firstly, the determination of the moderator effect of green self-identity on the effect of ecocentric and anthropocentric approaches, which are the sub-dimensions of the NEP, on the ecotourism attitude has been interpreted. In light of this information, it can be interpreted that the level of self-acceptance of the individual as a green consumer is important for the relationship between the values that are based on the environment and the values that keep the individual in the center, as well as the attitudes of individuals towards ecotourism activities. The rise of green identity in individuals will increase the effect of environment and individual-centered values on positive attitudes towards ecotourism. Because of the fact that the aforementioned NEP sub-dimensions increase the positive effect of the view towards ecotourism, it is thought that increasing the level of green self-identity in individuals will also be beneficial. Apart from this, the rejected moderator effect hypothesis should also be considered. The moderator role of green self-identity in the effect of balanced growth, which is one of the sub-dimensions of NEP, on ecotourism attitudes could not be determined. The most important point that causes this result is that balanced growth differs from the other two dimensions of the NEP in the matter of scope. If it must be discussed, while ecocentric and anthropocentric approaches describe the concept within the framework of the individual-environment, it is seen that the expressions measuring the balanced growth dimension focus on economic development and industrialization. In this context, the result obtained about the subject variable can explain the absence of a moderating effect due to the fact that the green identity of the consumer is individual-oriented. It is possible to come across studies examining the relationship with different variables, such as green self-identity and intentions to stay in green hotels [70], theory of planned behavior [70,119,120], and pro-environmental action [121]. Additionally, although there is a study in the literature determining the moderator role of green self-identity in the relationship between energy-efficient heating appliances and behavior change [122], no study has been conducted on the moderator role of self-identity in a model including other variables that are the subject of our research. From this point of view, the results of the study provide valuable findings to the tourism literature.

It can be understood that individuals are aware of the change in climatic conditions and global warming by comparing them with the past, with the determination of the effect of belief in climate change on all sub-dimensions of the NEP. It is observed that this awareness has an impact on the environment/human-centered values and environmental awareness and the idea of balanced growth to ensure sustainability. As individuals’ perception of the effects of changes in weather conditions and increasing drought escalate, their belief that the balance of nature with limited resources can easily be disturbed, that the industry should grow in a controlled manner, and that it is important for people to adapt without interfering with the natural environment increases. Although this result that has been obtained resembles the results of the research by Xue et al. [117], the absence of a study measuring the relationship between the mentioned variables in the tourism sector contributes to the originality of the research and to the literature. Moreover, although there are studies in which the effect of the sub-dimensions of NEP on the ecotourism attitude or nature-based tourism activities are discussed [61], it was concluded that there is not enough of them. Especially, since it measures the tourist perception in recent years, this study will shed light on the literature. In addition, when viewed as a whole, the study has the potential to make significant contributions to the literature within the framework of the model created and the associated variables.

### 6.2. Practical Implications

Understanding the link between climate change beliefs and eco-tourism attitudes can inform destination marketing and management strategies aimed at attracting environmentally conscious tourists. The study findings can be used to design educational and awareness-raising campaigns targeted at tourists to promote sustainable tourism practices. In this framework, the results of the research provide clues to the sector representatives in terms of the formation of tourists with high environmental awareness, bringing tourists who already have environmental awareness to the destination, and thus accelerating ecotourism activities. The role of green self-identity as a moderator of the relationship between climate change beliefs and eco-tourism attitudes can be used to target specific segments of the tourism market, such as individuals who identify strongly with environmental values. In addition, understanding the relationship between the NEP and eco-tourism attitudes can inform the development of policies and regulations aimed at promoting sustainable tourism practices.

It is important to plan the content education and teaching programs and activities in such a way that environmental awareness is established from the earliest stages of formal education, through undergraduate and postgraduate education, in order to increase the number of individuals with high environmental awareness. In this sense, it is essential for politicians and the education sector to take action on the subject during the curriculum-development stage. The accommodation businesses, which are the engine of the tourism sector, must take action in order to implement eco-labels and environmental management systems. In this regard, it is of great importance for the government to support and encourage the efforts of the businesses.

### 6.3. Limitations and Future Research Directions

As with any research, this study also had some limitations. One of them is that the study may have limited generalizability as it is based on a specific sample of tourists, which may not be representative of the broader population of tourists. Moreover, the study relied on self-reported data, which may be subject to social desirability bias.

Another limitation of the study is that the sample is particularly limited to Antalya/Alanya. Including different destinations in the research is important in order to generalize the effects of the climate crisis on tourism and to compare the results. In addition, the study could also measure environmental attitudes of different cultures within the framework of the variables considered. Additionally, the research model could be redesigned to make comparisons between hotels that have eco-label or environmental management systems and others, for future studies. This study has been carried out within the context of five-star hotel businesses, and this could be considered a limitation of the research. Future studies can be conducted based on different types of accommodation businesses and tourists, as well. 

This study also measured the perceptions and consequences of climate change among tourists visiting a specific destination. Similar themed studies with different methods with different sample groups can be conducted in future studies. For example, in order to determine the effects of climate change perceptions on the travel intentions of individuals who tend to visit a particular destination, an interview technique can be applied. Furthermore, the climate change beliefs of hotel business managers or destination management organizations can be evaluated through qualitative methods.

It is believed that with the increase in nature-based tourism activities in the post-COVID-19 period, there has been a considerable change in the environmental awareness and attitudes of individuals. In this context, new research models that will reveal the differences between the pre-pandemic and post-pandemic periods can be suggested for future studies. The study has been conducted based on tourists, who are one of the drivers of the tourism movement. Examining a similar model from the perspective of green hotel management using the same variables would also add value to the field in terms of results.

## Figures and Tables

**Figure 1 ijerph-20-04967-f001:**
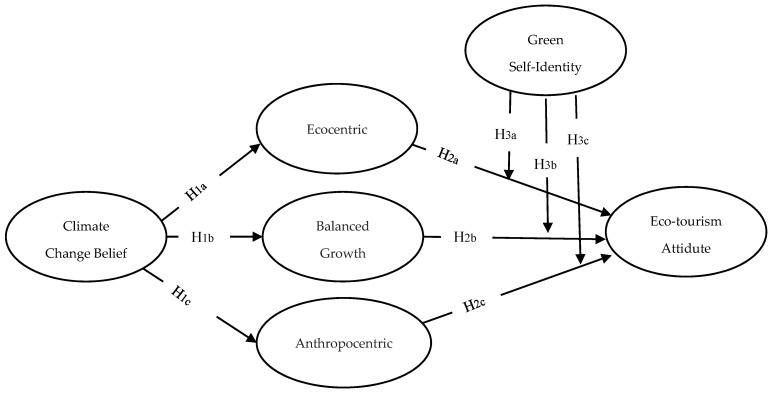
Proposed model.

**Figure 2 ijerph-20-04967-f002:**
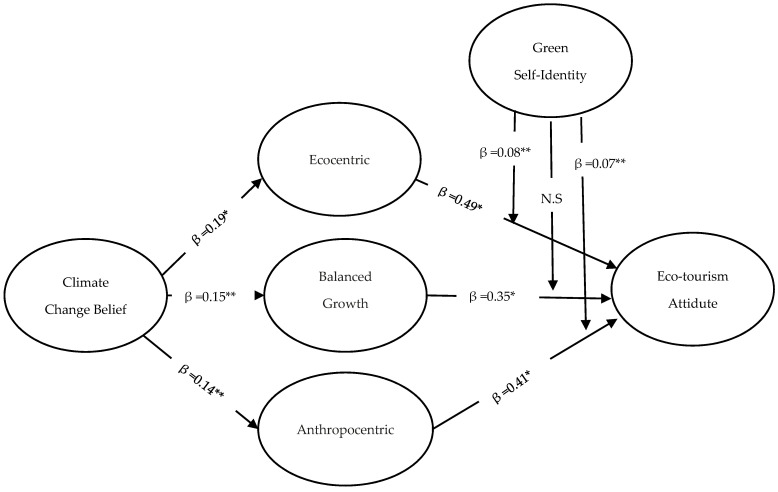
Estimates of structural equation modeling (SEM). * *p* < 0.001 ** *p* < 0.05 N.S.: No significant.

**Table 1 ijerph-20-04967-t001:** Summary and comparison of related studies in the field of climate change and NEP.

Author	Article Title	Method	Sample	Analysis	Conclusion
Dietz, T., Dan, A., Shwom, R. (2007) [29]	Support for Climate Change Policy: Social Psychological and Social Structural Influences	Quantitative analysis with online survey	316 respondents	Structural equation modeling (SEM), AMOS 5.0	The participants moved away from the understanding that ignores environmental factors. Respondents lost confidence in industrial companies and governments regarding environmental issues, while they placed more trust in environmental NGOs.
Ntanos, S., Kyriakopoulos, G., Skordoulis, M., Chalikias, M., Arabatzis, G. (2019) [30]	An Application of the New Environmental Paradigm (NEP) Scale in a Greek Context	Quantitative analysis with online survey	732 respondents	Spss: regression analyses	The answers given by the participants in the research show that environmental awareness spreads rapidly among individuals. Climate-based change and emerging threats have a significant impact on this. In addition, participants tend to learn about and use renewable energy sources.
Prober, S. M., Doerr, V. A., Broadhurst, L. M., Williams, K. J., Dickson, F. (2019) [31]	Shifting the conservation Paradigm: A Synthesis of Options for Renovating Nature under Climate Change	Literature Review	473 studies	Content analyses and typology	With climate change factors becoming a serious threat, there have been radical changes in environmental assessments, and this change has helped to create a new environmental life-building awareness in society.
Derdowski, L. A., Grahn, A.H., Hansen, H., Skeiseid, H. (2020) [32]	The New Ecological Paradigm, Pro-Environmental Behaviour, and the Moderating Effects of Locus of Control and Self-Construal	Quantitative analysis with online panel	200 respondents	Spss: regression analyses, PROCESS macro	According to the findings of the researchers, the new environmental paradigm shows that individuals evaluate seasonal and climate-based changes negatively, and therefore, they approach environmental issues in a more detailed, more careful, and more analytical way.
Diakakis, M., Skordoulis, M., Savvidou, E. (2021) [33]	The Relationships between Public Risk Perceptions of Climate Change, Environmental Sensitivity and Experience of Extreme Weather-Related Disasters: Evidence from Greece	Quantitative analysis with online survey	449 respondents	Spss: *t*-tests, correlation analysis and regression analysis	Participants of the research have better grasped the importance of environmental issues. In particular, as the degree of impact of natural disasters increased, the environmental awareness, practices, and approach sensitivities of individuals after the disaster increased rapidly.
Whitmarsh, L. (2011) [34]	Scepticism and uncertainty about climate change: Dimensions, determinants and change over time	Quantitative analysis: postal survey	551–589 respondents	SPSS: regression analysis	Findings show denial of climate change is less common than the perception that the issue has been exaggerated. Skepticism was found to be strongly determined by individuals’ environmental and political values (and indirectly by age, gender, location, and lifestyle), rather than by education or knowledge.
Whitmarsh, L. (2008) [35]	Are flood victims more concerned about climate change than other people? The role of direct experience in risk perception and behavioural response	Quantitative analysis: postal survey	589 respondents	SPSS: regression analysis Chi-square tests	Respondents are significantly more likely to consider climate change a salient risk and to take action in response to it. Therefore the relationship between air pollution experience and responses to climate change may be indirect and mediated by environmental values. The paper concludes by highlighting implications of this research for developing climate change policies and strategies for public engagement.
Yu, T., Linb, F., Kaob, K., Chaoc, C., Yud, T. (2019) [36]	An innovative environmental citizen behavior model: Recycling intention as climate change mitigation strategies	Quantitative analysis: face-to-face	514 respondents	Chi-square and ANOVA, structural equations modeling (SEM), partial least squares (PLS)	Environmental citizens whose recycling knowledge behavior is mature feel emotional. The path coefficients of the research model are connected to nature, which enhances their civic virtue (advocacy and activism) and arouses their recycling intention.

**Table 2 ijerph-20-04967-t002:** Summary and comparison of related studies in the field of NEP and eco-tourism.

Author	Article Title	Method	Sample	Analysis	Conclusion
Lee, W. H., Moscardo, G. (2005) [48]	Understanding the Impact of Ecotourism Resort Experiences on Tourists’ Environmental Attitudes and Behavioural Intentions	Quantitative analysis: face-to-face	242–396 respondents	SPSS: *t*-tests and chi-square, ANOVA	This study found few significant differences in respondents’ environmental awareness, attitudes, and preferences between the pre-visit and post-visit samples. Respondents in both samples strongly endorsed the principles of “intergenerational equity” and “harmony with nature”.
Ogunjinmi, A. A. (2016) [49]	Segmenting and Understanding Ecotourists in Nigeria National Parks by Environmental Attitudes	Quantitative analysis: face-to-face	350 respondents	SPSS: Chi-square, correlation analyses, multiple regression analysis	Based on the findings from the study, the identified segments are defined by different levels of environmental attitudes and could be further categorized based on different ranges of socio-demographic and trip characteristics that were significant in the regression models.
Formica, F., Uysal, M. (2001) [50]	Segmentation of Travelers Based on Environmental Attitudes	Quantitative analysis: mail and telephone survey	1120 respondents	Factor-cluster analysis, factor analysis, one way (ANOVA) analysis	According to the findings, trip behavior, destination attributes, and travelers’ choice of destination, rather than demographic characteristics, account for most of the variance among the three market segments: conservationists, anthropocentrics, and optimists.
Zhang, H., Lei, S. (2009) [51]	Residents’ environmental attitudes and behavioral intention of tourism development in Beimen Coastal Wetland area, Taiwan	Quantitative analysis: on-site interviews	361 respondents	One-way analysis of variance, post-hoc, two-stage cluster analysis, K-means cluster analysis	Three types of respondents including strong, moderate, or weak environmental attitudes toward tourism development were identified. Furthermore, three types of respondents differenced in behavioral intention of tourism development.
Jeong, E., Lee, T., Brown, A.D., Choi, S., Son, M. (2021) [52]	Does a National Park Enhance the Environment-Friendliness of Tourists as an Ecotourism Destination?	Quantitative analysis: face-to-face	952 respondents with online survey	SPSS: factor analysis, AMOS	The results suggested the necessity of national parks to expand educational programs and facilities for eco-tourists visiting national parks, maintaining a balanced relationship between themselves and nature, and having a strong environmental awareness.
Jackson, S. (2007) [53]	Attitudes Towards the Environment and Ecotourism of Stakeholders in the UK Tourism Industry with Particular Reference to Ornithological Tour Operators	Quantitative analysis: face-to-face	165 respondents	SPSS: factor analysis, regression analysis	Significant correlations were measured between the NEP and TES scales suggesting that general attitudes affect specific attitudes. No significant relationships were established between attitudes and stated or implied behavior based on the tour operators’ brochures.
Khanh, C. N. T., Phong, L. T. (2020) [54]	Impact of environmental belief and nature-based destination image on ecotourism attitude	Quantitative analysis: face-to-face	479 respondents	SPSS: correlation analysis, structural equation modeling (SEM)	This study found statistically significant and positive effects of time perspectives, environmental beliefs, and nature-based destination images on ecotourism attitudes. The results also indicate that environmental belief is found to have statistically significant effects, both direct and indirect, on ecotourism attitudes; its indirect effect is transmitted through a nature-based destination image.

**Table 3 ijerph-20-04967-t003:** Summary and comparison of related studies in the field of green self-identity.

Author	Article Title	Method	Sample	Analysis	Conclusion
Sharma, N., Lal, M., Goel, P., Sharma, A., Rana, N. P. (2022) [66]	Being socially responsible: How green self-identity and locus of control impact green purchasing intentions?	Quantitative analysis with online survey	391 Indian customers	PLS-SEM, Importance Performance Map Analysis (IPMA)	The results show a partial sequential mediation between green self-identity and green purchase intention.
Whitmarsh, L., O’Neill, S. (2010) [67]	Green identity, green living? The role of pro-environmental self-identity in determining consistency across diverse pro-environmental behaviors.	Quantitative analysis with postal survey	551 respondents	SPSS: regression analyses	Pro-environmental self-identity is an important predictor for some other pro-environmental behaviors.
Hui, Z., & Khan, A. N. (2022) [68]	Beyond pro-environmental consumerism: role of social exclusion and green self-identity in green product consumption intentions.	Quantitative analysis with online survey	476 respondents	Structural equation modeling (SEM) method with AMOS-24	The interaction between social exclusion and green self-identity is significant for green product attitudes and subjective norms regarding green products.
Becerra, E. P., Carrete, L., Arroyo, P. (2023) [69]	A study of the antecedents and effects of green self-identity on green behavioral intentions of young adults.	Quantitative analysis with online survey	298 respondents	Structural equation model (PLS-SEM) using SmartPLS-3 and post-hoc analysis	Green self-identity and green product values positively affect behavioral intentions.
Fatoki, O. (2020) [70]	Consumers’ intention To Stay In Green Hotels In South Africa: The Effect Of Altruism And Green Self-Identity.	Quantitative analysis with face-to-face	416 respondents	Structural equation modeling (PLS SEM)	There are significant positive relations among descriptive norms, perceived behavioral control, altruism, green self-identity, and the intention of consumers to visit green hotels.
Silintowe, Y. B. R., Sukresna, I. M. (2022) [71]	Green Self-Identity as a Mediating Variable of Green Knowledge and Green Purchase Behavior	Quantitative analysis with face-to-face	271 respondents	PLS-SEM approach with SmartPLS Version 3.0	Green self-identity affects purchase behavior and is a mediator between knowledge and purchase behavior.
Qamar, R., Abidin, Z. U., Fayyaz, S., Ahmad, H. (2022) [72]	The influence of green human resource management practices on employee’s green creativity: The roles of green self-identity & green shared vision	Quantitative analysis with postal survey	202 respondents	PLS-SEM approach with SmartPLS	Green human resource management practices have a significant effect on the green creativity of employees, and green self-identity mediates this effect.
Siddiquei, A., Asmi, F., Asadullah, M. A., Mir, F. (2021) [73]	Environmental-specific servant leadership as a strategic tool to accomplish environmental performance: a case of China.	Quantitative analysis with face-to-face	125 respondents	Multi-level modeling (MLM) and ordinary least squares (OLS) regression	It was shown that environmentally specific servant leadership has a negative effect on green self-identity, which in turn predicts subsequent pro-environmental behavior.
Vesely, S., Masson, T., Chokrai, P., Becker, A. M., Fritsche, I., Klöckner, C. A., Panno, A. (2021) [74]	Climate change action as a project of identity: Eight meta-analyses.	Literature research	188 articles	Meta-analyses	The results show that there are robust, medium-sized, and strong connections between both pro-environmental intentions and behaviors with people’s commitment to nature, environmental self-identity, and identification with groups thought to support climate-friendly behaviors.
Shadiqi, M. A., Djuwita, R., Febriana, S. K. T., Septiannisa, L., Wildi, M., Rahmawati, Y. (2022) [75]	Environmental Self-Identity and Pro-Environmental Behavior in Climate Change Issue: Mediation Effect of Belief in Global Warming and Guilty Feeling	Quantitative analysis with face-to-face	202 respondents	Mediation regression analysis with Model 4 of PROCESS	The mediator model analysis has shown that the global warming belief partially mediates the relationship between environmental self-identity and PEB, and the sense of guilt is not a significant mediator.

**Table 4 ijerph-20-04967-t004:** Structural model confirmatory factor analysis results.

Factors/Items	Standard Loadings	t-Value	*R* ^2^	CR	AVE	CA
**Climate change belief**				0.954	0.751	0.955
I believe that weather conditions have changed (precipitation and temperature) compared to the past.	0.862		0.74			
I believe there has been a decrease in snow and rain compared to the past.	0.848	24.28 *	0.72			
I believe that more drought, dust, and other unusual weather events have occurred in recent years.	0.844	23.54 *	0.71			
I believe that the dry season in recent years comes sooner than in the past.	0.900	23.35 *	0.81			
I believe that winter here is not as cold as it was in the past.	0.847	26.51 *	0.71			
I believe that winds are particularly strong in summer and stir up dust.	0.898	23.46 *	0.80			
I am sure global warming is taking place.	0.867	26.36 *	0.75			
**New environmental paradigm**						
**Ecocentric**				0.934	0.739	0.933
We are approaching the limit of the number of people the Earth can support.	0.804		0.64			
The Earth is like a spaceship with only limited room and resources.	0.855	20.62 *	0.73			
The balance of nature is very delicate and easily upset.	0.896	22.06 *	0.80			
When humans interfere with nature, it often produces disastrous consequences.	0.859	20.74 *	0.73			
Mankind is severely abusing the environment.	0.883	21.60 *	0.78			
**Balanced Growth**				0.847	0.649	0.876
To maintain a healthy economy, we will have to develop a “steady state” economy where industrial growth is controlled.	0.837	19.18 *	0.70			
There are limits to growth beyond which our industrialized society cannot expand.	0.825	18.82 *	0.68			
Humans must live in harmony with nature to survive.	0.754		0.72			
**Anthropocentric**				0.838	0.662	0.886
Humans do not have the right to modify the natural environment.	0.768		0.59			
Humankind was not created to rule over the rest of nature.	0.830	17.55 *	0.68			
Plants and animals do not exist primarily to be used by humans.	0.854	18.08 *	0.73			
Humans need to adapt to the natural environment because they cannot remake it to suit their needs	0.800	16.85 *	0.63			
**Eco-tourism attitude**				0.880	0.605	0.845
Tourism in sustainably managed tourist areas should avoid interfering with the habitat of local flora and wildlife.	0.752	11.87 *	0.56			
The role of sustainably managed tourist areas goes beyond their economic function.	0.523	9.10 *	0.27			
Sustainable tourism can enhance visitors’ personal development.	0.808	12.41 *	0.65			
Visiting sustainably managed tourist areas should be subject to a higher relative payment.	0.871	12.92 *	0.75			
Tourism in sustainably managed tourist areas should restrict visits to preserve important cultural values and norms.	0.712	11.45 *	0.50			
Part of the income from tourism should fund the promotion of environmental and cultural conservation.	0.591		0.34			
**Green self-identity**				0.854	0.661	0.911
I think of myself as someone who is very concerned with environmental issues.	0.809		0.65			
Being environmentally friendly is an important part of who I am.	0.823	22.45 *	0.85			
I identify with the aims of environmental groups like Greenpeace, Friends of the Earth, etc.	0.808	22.16 *	0.82			

* *p* < 0.001.

**Table 5 ijerph-20-04967-t005:** Bootstrap regression analysis results.

**H_3a_:**					Eco-Tourism Attitude		
					β	Confidence Interval	
						Lower limit	Upper limit
Ecocentric (X)					0.25 **	0.065	0.585
Green self-identity (W)					0.27 **	0.054	0.602
X.W (Interaction)					0.08 **	0.001	0.156
R^2^					0.23		
Green self-identity	β	S.E.	t	LLCI	ULCI		
Low	0.49 *	0.06	7.61	0.368	0.624		
Middle	0.60 *	0.05	9.10	0.495	0.708		
High	0.65 *	0.06	9.92	0.524	0.784		
**H_3b_:**					Eco-tourism attitude		
					β	Confidence Interval	
						Lower limit	Upper limit
Balanced growth (X)					−0.10 *^a^*	−0.338	0.121
Green self-identity (W)					−0.07 *^a^*	−0.259	0.108
X.W (Interaction)					0.02 *^a^*	−0.026	0.083
**H_3c_:**					Eco-tourism attitude		
					β	Confidence Interval	
						Lower limit	Upper limit
Anthropocentric (X)					0.18 **	0.139	0.255
Green self-identity (W)					0.25 **	0.027	0.483
X.W (Interaction)					0.07 **	0.008	0.129
R^2^					0.20		
Daily green behavior	β	S.E.	t	LLCI	ULCI		
Low	0.19 *	0.04	4.09	0.104	0.295		
Middle	0.29 *	0.04	6.73	0.206	0.377		
High	0.33 *	0.05	6.28	0.232	0.444		

* *p* < 0.001, ** *p* < 0.05, *^a^*: No significant.

## Data Availability

The data analyzed during this study are available on request from the corresponding author.

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
