# Peer review of "Effect of Climate Change Belief and the New Environmental Paradigm (NEP) on Eco-Tourism Attitudes of Tourists: Moderator Role of Green Self-Identity"

_ijerph, 2023, doi:10.3390/ijerph20064967_

Round 1

Reviewer 1 Report

The definition of eco-tourism was not adequate, and as a result, the connection to green self-identity was a bit unclear. The authors might want to clarify the relationship between eco-tourism and green self-identity for the readers' sake.

The survey is extensive and is statistically persuasive. The overall merit is good, as the study seems to aim accurately at locating ways to change behaviors toward a greener, eco-friendly norm.

Author Response

Thank you very much for your precious comments and support. The manuscript has been revised and all the necessary changes have been made in accordance with your suggestions. 

Reviewer 1

Thank you so much for your positive comments and assessment of our topic. Your comments and suggestions have been very helpful for further improving of our paper.

We greatly appreciate your feedback.

The changes which were made about the manuscript have been highlighted in our text.

Comment

Revision

The definition of eco-tourism was not adequate, and as a result, the connection to green self-identity was a bit unclear. The authors might want to clarify the relationship between eco-tourism and green self-identity for the readers' sake.

Thank you very much for your valuable comment. In the light of your suggestions, more studies on ecotourism have been cited in the article. In addition, its relationship with different disciplines has also been evaluated (Page 2-line 61-81;  Page 7-Table 2; Page 8-line 281-293).

Comment

Revision

The survey is extensive and is statistically persuasive. The overall merit is good, as the study seems to aim accurately at locating ways to change behaviors toward a greener, eco-friendly norm.

Thank you very much for your positive comment. We would like to state that your nice comments encourage us even more for future studies.

Reviewer 2 Report

Dear authors

The topic of your research is interesting.

The main problem with the research is that it is very limited to a very small part of the population. As I understand, you have only reached people who come to a specific region, and moreover to specific 5-star hotels. This can be beneficial for certain studies and research questions related to behavioral observations or to specific contextual features. But that is not the subject of your inquiry. Why did you need this specific context for your research? If you are interested in this context, reformulate your RQs and the scope of your research.

To test your model, or to test your hypotheses, you could have interviewed a large database, even consisting of respondents who did not visit a specific place. You could have asked them some control questions about their travel behavior.

The introduction is very general and does not sufficiently introduce tourism or the specific region.

Perhaps it is a better idea to first describe NEP and ecotourism and arrive at these hypotheses and then describe the other antecedents. Now it will take you some time to define your main constructions.

When building your hypotheses, you give us some nice and some less relevant references. But you just mention the research and don't really build on why they are indicative of your hypotheses. It may be an idea to make a table with the relevant variables and outcomes of the studies.

In the description of the context you write that some segments may be more aware of the environment/climate. You describe farmers. How does this relate to your research.

Sometimes your description of concepts is a bit confusing: e.g., in short, green self-identity arises when individuals think deeply about environmental issues and shape their practices in their lives accordingly. This is not an appropriate definition of GI. This could be GI due to…

also in the conclusion e.g.: In the light of this information, it can be interpreted that the level of self-acceptance of the individual as a green consumer is important in the relationship between the values ​​based on the environment and those that place the individual at the center and the attitude of individuals towards ecotourism activities.

Have you said anything about ID acceptance before, have you measured it?

The questionnaire is simple but okay, apart from missing some control variables. The analyzes are well done. The presentation is a bit over the top. One large table with the global effects and the effects depending on the green identity can provide a good overview.

You can also provide the reader with a drawing of the model, including the H's.

Author Response

Thank you very much for your precious comments and support. The manuscript has been revised and all the necessary changes have been made in accordance with your suggestions. 

Reviewer 2

Thank you so much for your positive ideas and assessment of our topic. Your comments and suggestions have been very helpful for further improving of our paper.

We greatly appreciate your feedback.

The changes which have been made regarding the manuscript have been highlighted in our text.

Comment

Revision

The main problem with the research is that it is very limited to a very small part of the population. As I understand, you have only reached people who come to a specific region, and moreover to specific 5-star hotels. This can be beneficial for certain studies and research questions related to behavioral observations or to specific contextual features. But that is not the subject of your inquiry.

Thank you very much for your valuable comment, which strengthens our article. Based on your comment, necessary additions have been made and presented in the sampling and data collection section (Page 12-line 426-432 and 441-446).

Comment

Revision

To test your model, or to test your hypotheses, you could have interviewed a large database, even consisting of respondents who did not visit a specific place. You could have asked them some control questions about their travel behavior.

Thank you very much for your contribution to our study. In this context, it has been presented as a suggestion in the limitation and future research section of the study (Page 19-line 699-705).

Comment

Revision

The introduction is very general and does not sufficiently introduce tourism or the specific region .Perhaps it is a better idea to first describe NEP and ecotourism and arrive at these hypotheses and then describe the other antecedents. Now it will take you some time to define your main constructions.

Thank you very much for your valuable contribution. Throughout the study, the changes have been highlighted by making additions in line with your suggestion.

Comment

Revision

When building your hypotheses, you give us some nice and some less relevant references. But you just mention the research and don't really build on why they are indicative of your hypotheses. It may be an idea to make a table with the relevant variables and outcomes of the studies.

I would like to state that we found your precious comment to be justified. In this context, we have examined the studies in the literature on variables and presented them in our study in the form of a table. (Page 7-Table2; Page 10-Table 3).

Comment

Revision

In the description of the context you write that some segments may be more aware of the environment/climate. You describe farmers. How does this relate to your research.

Sometimes your description of concepts is a bit confusing: e.g., in short, green self-identity arises when individuals think deeply about environmental issues and shape their practices in their lives accordingly. This is not an appropriate definition of GI. This could be GI due to…

We would like to state that we find your comment valuable and justified. In this context, the necessary revision has been made and highlighted.  In the introduction part, the relevant part has been removed and the sentences have been reconstructed (Page 2-3- Line 61-90).

Comment

Revision

also in the conclusion e.g.: In the light of this information, it can be interpreted that the level of self-acceptance of the individual as a green consumer is important in the relationship between the values ​​based on the environment and those that place the individual at the center and the attitude of individuals towards ecotourism activities.

Have you said anything about ID acceptance before, have you measured it?

The questionnaire is simple but okay, apart from missing some control variables. The analyzes are well done. The presentation is a bit over the top. One large table with the global effects and the effects depending on the green identity can provide a good overview.

You can also provide the reader with a drawing of the model, including the H's.

Thank you very much for your valuable statements.  In this context, necessary revisions were made and highlighted. In addition, a hypothetical research model, in which both path coefficients and hypotheses are included, has been added (Page 11-Figure 1;  Page 11-Figure 2).

Reviewer 3 Report

in general the research is good, but it is necessary to add a more specific explanation of the purpose and contribution, and at the conclusion of this research must be able to answer the purpose and contribution

Author Response

Thank you very much for your precious comments and support. The manuscript has been revised and all the necessary changes have been made in accordance with your suggestions. 

Reviewer 3

Thank you so much for your positive comments and assessment of our topic. Your comments and suggestions have been very helpful for further improving of our paper.

We greatly appreciate your feedback.

The changes which were made about the manuscript have been highlighted in our text.

Comment

Revision

in general the research is good, but it is necessary to add a more specific explanation of the purpose and contribution, and at the conclusion of this research must be able to answer the purpose and contribution

Thank you very much for your valuable contribution to our work. In line with your suggestions, necessary changes have been made in the introduction section by giving different references. In addition, new recommendations have been developed in the limitation and future research part of the study (Page 2-3-Line 61-90;  Page 19- Line 699-705).

Round 2

Reviewer 2 Report

Dear Authors

appreciate the figures you now put in the text. That's clearer. You also made some additional changes. Thank you!

Although you motivate why you only conducted the research at a five-star resort, this remains a serious weakness of your research. Either you take this scope and adjust your literature and introduction related to luxury tourism, or you explain that you are targeting a specific segment of the population. You might even wonder why people stay at this hotel: business/ holidays..

OR

You can add another data set; more representative of the tourism industry

Unfortunately, I think it is very sensitive to do a survey on this region at this time, because of the tragedy in this beautiful country. So you cannot expand your sample with people from other hotels in this region (quota sample) etc. to make your research more represernatative.

So you could consider doing a second wave  of the study, with a more representative sample for all kinds of tourists. The destination does not have to be this specific region, as your dependent variable is the attitude towards ecotourism. You can ask a large sample of respondents from many regions to complete this fundamental questions. You can even ask additional control questions about the last travel destination, the frequency of travelling, the main reasons for travelling... You can choose these control variables related to numbers you may find (eg. studies on Internet)  about eg. reasons for traveling, frequency of traveling... and see the comparison with these numbers as indicative for your sample accurancy. 

Author Response

Dear Professor,

We have made the necessary changes and additiıons to the manuscript in accordance with your suggestions and uploaded the report to the system.

Yours sincerely,
